# Successful Management of Spondylodiscitis Caused by *Aspergillus nidulans*: A Case Report and Literature Review

**DOI:** 10.3390/jof11050361

**Published:** 2025-05-06

**Authors:** Safia Nadji, Nicolas Ettahar, Jordan Leroy, Gisèle Dewulf, Edith Mazars

**Affiliations:** 1Laboratoire de Biologie Médicale, Microbiologie, CH de Douai, 59300 Douai, France; safia.nadji@ch-douai.fr; 2Service de Maladies Infectieuses, CH de Valenciennes, 59322 Valenciennes, France; ettahar-n@ch-valenciennes.fr; 3Service de Parasitologie-Mycologie, CHU Lille, 75013 Lille, France; jordan.leroy@chru-lille.fr; 4Laboratoire de Biologie Médicale, Microbiologie, CH de Valenciennes, Avenue Désandrouin, 59300 Valenciennes, France; dewulf-g@ch-valenciennes.fr

**Keywords:** spondylodiscitis, *Aspergillus nidulans*, immunocompromised

## Abstract

We report a case of spondylodiscitis caused by *Aspergillus nidulans* (SC*AN*). *A. nidulans* is a saprophytic fungus and emerging pathogen responsible for a variety of infections, although it is rarely implicated in osteoarticular infections. The patient was a 59-year-old immunocompromised patient with a history of lymphoma and splenectomy. Following diagnosis, the patient was promptly and effectively treated with voriconazole. A literature review underlines the distinctive features of the few case reports of SC*AN*, as well as the original features of the present case report.

## 1. Introduction

*Aspergillus nidulans* is a filamentous saprophytic fungus in the phylum Ascomycota. In immunocompromised patients, it can lead to sinusitis and pulmonary infections. *A. nidulans* can be distinguished from other *Aspergillus* species by its dark green colonies, which appear brown on the reverse side, its short, brown conidiophores, and its biseriate heads [1].

Osteoarticular infections caused by the *Aspergillus* spp. are rare and mainly affect immunocompromised patients [2,3]. The most commonly reported species in this context is *A. fumigatus* (55%), followed by *A. flavus* (12%) and then *A. nidulans* (7%) [3]. We describe a case of spondylodiscitis caused by *A. nidulans* (SC*AN*) in an immunocompromised patient with a history of lymphoma and splenectomy.

## 2. Case Report

On 26 September 2018, a 59-year-old man was seen in consultation in our Infectious Disease Department for the follow-up of a left pneumococcal empyema and endocarditis caused by *Streptococcus pneumoniae* in June 2018, for which the patient was treated with amoxicillin for 6 weeks. His medical history included hypothyroidism, viral pericarditis in 1998, bacteremia due to *Streptococcus gallolyticus* in 2016, ulcerative esophagitis, and non-Hodgkin lymphoma (NHL). During the consultation, he reported suffering from lower back pain for 2 weeks. The lumbar pain was associated with paresthesia of the lower left limb. Magnetic Resonance Imaging (MRI) was performed a week later and showed edema associated with erosion of the L1–L2 vertebral endplates (Figure 1), suggestive of an intravertebral disc herniation or infectious spondylodiscitis. Secondary MRI was performed 10 days later and showed persistence of the previous images.

The patient was hospitalized in the Infectious Disease Department on October 25 to perform disc biopsies, respecting the 2-month therapeutic window following his antibiotic treatment. His laboratory results included a blood leukocyte count of 8.9 × 10^9^/L, with 74% neutrophils, and a C-reactive protein level of 7.7 mg/L. Two disc biopsies were performed and sent to the microbiology and pathology laboratories for analysis. Five days of culture at 35 °C on bacterial culture media (Columbia Selective Agar with Sheep Blood and Chocolate agar with Vitox, Thermo Scientific^®^, France) and on Sabouraud plates (Sabouraud agar with gentamicin and chloramphenicol, Thermo Scientific^®^) yielded green colonies in pure culture. A direct examination with methylene blue was performed and showed short conidiophores prolonged with small vesicles. The *Aspergillus* heads were biseriate, in flared columns, and the microconidia were echinulate. As there were no Hulle cells or cleistothecia, the *Aspergillus* spp. was identified, and the culture was sent to the Mycology Department at the Regional University Hospital. The culture strain was analyzed via conventional PCR targeting the internal transcribed spacer (ITS) region, using the primers ITS1 (5′-TCCGATAGGTGAACCTGCGG-3′) and ITS4 (5′-TCCTCCGCTTATTGATATGC-3′), followed by sequencing of the 375 bp amplification products. Taq polymerase (Taq Hot Diamond, Eurogentec) and TaqMan polymerase were employed for the amplification of the target fragment (<500 bp) and for sequencing, respectively. BLAST analysis (Version 2.8.1) revealed a 99.2% nucleotide sequence identity with the sequence UOA/HCPF 9011 from the *A. nidulans* strain MITS 265 ISHAM-ITS. In parallel, we performed a galactomannan antigenemia assay (Platelia^TM^ Aspergillus Ag; Biorad^®^), which was positive at 0.83 (controlled at 1.1), and a (1-3)-β-D-glucan assay (Fungitell^®^), which was negative (48 pg/mL). Histopathologic examination of the biopsies revealed no evidence of fungal invasion or fungal elements.

Based on the clinical signs, imaging investigations, culture examination, and positive antigenic tests, a diagnosis of SCAN was made. An antifungal susceptibility test (AFST) was performed using the E-test (bioMerieux^®^) to determine the minimum inhibitory concentration (MIC), interpreted according to the breakpoints set by the European Committee on Antimicrobial Susceptibility Testing (EUCAST). The strain was susceptible to itraconazole and isavuconazole (Table 1).

There were no established breakpoints for voriconazole, posaconazole, caspofungin, or amphotericin B in 2018. Conventionally , in the absence of an interpretative threshold, a low MIC value is considered susceptible. The patient received 6 mg/kg of voriconazole every 12 h on day 1 followed by 4 mg/kg (350 mg) twice daily. Monitoring performed on day 5 of treatment found that the plasma voriconazole concentration was too high at 10.35 mg/L (trough levels between 1 and 5 mg/L), so the dose was reduced to 200 mg twice daily. The plasma concentrations of voriconazole were monitored on the 12th and 20th days, with the levels measured at 5.69 and 2.98 mg/L, respectively. Spine immobilization was prescribed, but no surgery was required. As recommended by the National Reference Center of Mycology, the patient received antifungal treatment with voriconazole for 6 months, with clinical improvement observed after 3 months (from January) and radiologic regression a month later. The center advised discontinuing the therapy based on clinical and imaging findings.

The galactomannan serum antigen was measured twice in November and at every follow-up consultation (in January, February, and April). It remained positive for 3 months until January. To understand the route of infection, fibroscopy and bronchoalveolar lavage were performed, revealing an inflammatory aspect of the bronchi. The culture examination from the fibro-aspiration found a colony of *A. fumigatus* and the development of *Candida albicans*. Additionally, sinus, cerebral, and pulmonary computer tomography (CT) scans were performed, but no microbiological documentation was found. Hepatic MRI found a lesion suggestive of a fungal abscess, but it was not explored further, as it was not accessible by ultrasound. A positron emission tomography (PET) scan performed in March found no abnormality. Thus, SC*AN* was considered successfully managed. Nevertheless, the patient died from comorbidities 4 years later, in October 2022.

## 3. Discussion

In the case we present here, the patient suffered from lower back pain associated with an inflammatory syndrome. MRI revealed vertebral lesions and microbiology results confirmed *A. nidulans* on two disc biopsies. These elements confirmed a diagnosis of SC*AN*. Diagnosis was further supported by galactomannan antigen testing, which was >0.8 in November and remained positive (>0.5) for 3 months after the initiation of antifungal therapy.


A limitation in the strain identification level should be noted. The molecular target (ITS) confirmed the *nidulans* complex. For species-level confirmation, markers like β-tubulin or calmodulin should be used [2].


It is important to remember that spondylodiscitis, or vertebral osteomyelitis, is an infection of the intervertebral disc, with a possible extension to adjacent vertebrae. Immunocompromised patients presenting inflammatory spinal pain associated with positive imaging should be investigated for infectious spondylodiscitis [4]. *Aspergillus* spondylodiscitis is diagnosed on a positive culture of bone tissue obtained via disc biopsy [5].

A review of the literature including the key words “spondylodiscitis” or “vertebral osteomyelitis”, “case report”, and “*Aspergillus nidulans*” was performed in January 2024 and found two published cases. In 2013, a case report and literature review by Jiang et al. [6] covering the period 1965 to 2012 found three cases of SC*AN*. In 2014, Gamaletsou et al. [7] analyzed 180 well-described cases of *Aspergillus* osteomyelitis over a period from 1947 to 2013 and also found three cases due to *A. nidulans*. Seven years later, Pernat et al. [8] found 112 cases of vertebral aspergillosis, with 2 being due to *A. nidulans*. The same year, Koutserimpas et al. [9] reviewed 118 cases of spondylodiscitis caused by *Aspergillus* spp. (SC*A*S) between 1936 and 2021, finding 7 due to *A. nidulans*. We summarize the findings of the 13 published case reports of SC*AN* in Table 2 [6,10,11,12,13,14,15,16,17,18,19]. Whatever the exact number of cases, SC*A*S is rare, especially cases due to *A. nidulans*.

The most recent published case report with *A. nidulans* described infection in an immunocompetent patient [18]. More generally, osteomyelitis caused by the *Aspergillus* spp. affects immunocompromised patients [3,6,7,9,20]. The main risk factors are corticosteroid use, primary immunodeficiencies, mainly chronic granulomatous disease (CGD), neutropenia, transplantation, hematological malignancy, solid tumors, HIV infection, and diabetes mellitus [3,20]. More specifically, *A. nidulans*, an extremely rare mold in patients with cancer, is isolated much more frequently in CGD patients and, indeed, with the same frequency as that of *A. fumigatus* [20]. On analyzing the thirteen published cases of SC*AN*, ten (77%) patients were immunocompromised, and nine of these (69%) were due to CGD. The male sex and young age of the majority of patients with SCAN are linked to CGD. Three patients were identified as immunocompetent, although the first case was probably unknown CGD. Our case is the first SC*AN* to be declared in an immunocompromised patient by a pathology other than CGD.

The most recent review of SC*A*S [9] reports that the lumbar spine is frequently affected (37.3%), as described in the present case report. However, this is not true for the 13 case reports of SC*AN* (Table 2) where the thoracic vertebra was the main localization for 12 (86%) and extensive in 5 (36%).

The definite diagnosis of SC*A*S remains delicate. Since 1994, most cases are suggested by lesions detected on MRI and diagnosed through cultures (73.7%), followed by histopathology, but rarely by the galactomannan antigen (2 cases = 1.8%) [9]. Moreover, antifungal susceptibilities are generally not provided [8,9]. In our case, the diagnosis was suspected on MRI images, and the involvement of the *Aspergillus* spp. was confirmed by cultures on two biopsies. The *A. nidulans* complex was identified through molecular biology, along with persistently positive galactomannan, and susceptibility tests were performed.

Although most of the authors do not describe the gateway [3,6,7,8,21], SC*A*S could be explained by three pathogenic mechanisms: direct invasion by contiguous pulmonary foci, hematogenous diffusion, and iatrogenic or traumatic inoculation [9,22]. In our case, fibroscopy and bronchoalveolar lavage did not provide contributive findings. Additionally, sinus, cerebral, and pulmonary CT scans revealed no abnormalities. Once again, the route of infection remained unidentified.

In 2023, an extensive systematic review analyzed 1072 patients with fungal osteomyelitis by *Aspergillus* (26.5%), followed by *Candida* (20.7%) and *Mucor* (16.8%) [23]. Full recovery was observed in 72.8%. These authors reported that *Aspergillus* infection was strongly associated with mortality, and vertebral involvement further reduced the chances of survival. Surprisingly, the three reviews focusing specifically on SC*A*S found quite similar percentages of positive outcomes: 71%, 71.2%, and 78.8% [3,8,9]. The mortality rate of SCAN appears to be high, with potential differences between *A. nidulans* osteomyelitis and spondylodiscitis, despite the small cohort sizes. Indeed, in 2004, *A. nidulans* osteomyelitis was associated with a 50% mortality (7/14 cases) compared to *A. fumigatus* osteomyelitis (0/10 cases) in CGD patients [24]. Ten years later, our review confirms these data with six (46%) patients who died (Table 2). Dotis et al. [24] specified that a successful outcome was achieved by prompt antifungal treatment combined with surgery and immunotherapy. In our case, SC*AN* was successfully resolved, more than likely because of prompt treatment on symptom onset, a non-extensive vertebra lesion, and weakening with the use of voriconazole.

Surgery to treat SC*A*S does not appear to be of added value to antifungal therapy for some authors [3,9,20], while it does for others [7,18]. More precisely, surgery was performed in 11 (85%) of the SC*AN* cases we reported (Table 2), probably due to the severity of the cases, but also due to non-compliant discontinuation of voriconazole after hospital discharge in one case. Surgical debridement was not necessary for the case we report here.

There is currently a lack of firm treatment recommendations for *Aspergillus* spondylodiscitis [25,26]. In the 2014 review by Gabrielli et al. [3], the most commonly used drug was amphotericin B, followed by itraconazole and voriconazole. In the first 2021 review, Koutserimpas et al. [9] report that voriconazole is used as the first antifungal agent (62%) after 2003. However, this contrasts with the second review published in 2021 by Perna et al. [8], which reports that medical treatments consisted mainly of a combination of two or more drugs (50 patients = 45%, 2 with voriconazole), with only 12 patients (10.8%) receiving voriconazole alone. Focusing on the thirteen cases of SC*AN*, nine (69%) and seven (54%) patients received amphotericin B or voriconazole alone or in combination, respectively (Table 2). The choice of the antifungal obviously depends on the year of the observation. In our case, voriconazole was selected as the drug of choice for invasive aspergillosis [20,27]. With reference to the data found in the literature, the duration of antifungal treatment seems to be at least 6 months. We consulted the National Reference Center of Mycology for advice in our case, who recommended 6 months of voriconazole and to check MRI and clinical values before stopping treatment. Notably, the recent 2023 systematic review of reported cases of fungal osteomyelitis found an association between longer treatment duration and increased survival, whereas shorter treatment was linked to higher mortality [23].

Some authors underline the importance of identifying the species [27] notably involved in SC*A*S [9]. The same authors emphasized the critical importance of performing antifungal susceptibility testing, as an increased rate of azole resistance was reported recently in the Netherlands and the United Kingdom [9]. Susceptibility tests are essential not only to treat the patient but also to monitor resistance to the antifungal. An example was described in the case report and review published in 2013 [6]. The patient underwent decompressive surgery before receiving voriconazole. Nevertheless, she did not continue voriconazole treatment after discharge and relapsed 16 months later. After the second drug susceptibility results, she continued with voriconazole treatment for a further 6 months with a successful outcome.

In 2016, the Infectious Diseases Society of America recommended that for the diagnosis and management of aspergillosis, serial monitoring of serum galactomannan could be used in appropriate patient subpopulations (hematologic malignancy, after hematopoietic stem cell transplantation) who have an elevated galactomannan level at baseline to monitor disease progression and therapeutic response and to predict the outcome [26]. Focusing specifically on SC*A*S, the role of serum galactomannan remains a topic of debate Ref. [8] and is rarely used as a diagnostic tool [8,9,23]. In our case, the galactomannan assay remained positive for 3 months. It appears, therefore, to be an interesting marker for suspected spondylodiscitis of fungal origin and constitutes another diagnostic element.

In our case, the (1-3)-β-D-glucan assay was not contributory in advancing the diagnosis of SC*AN*. This result supports the limited role of this marker in invasive aspergillosis [27].

This case represents the comprehensive management of a SCAN case, detailing the clinical signs, imaging findings, fungal isolation and identification, antifungal susceptibility, and galactomannan antigen dynamics. Based on our current knowledge, it is the first reported case of SCAN in an immunocompromised patient with a malignancy other than CGD to achieve a successful therapeutic outcome without surgical debridement.

## Figures and Tables

**Figure 1 jof-11-00361-f001:**
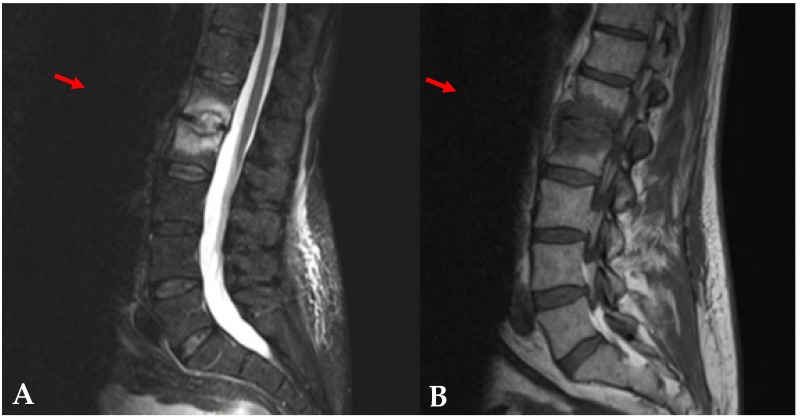
Magnetic Resonance Imaging of the patient. The sagittal STIR image (**A**) showed bright high-signal intensity of L1–L2 disc, which was well enhanced on the sagittal T1-weighted image (**B**).

**Table 1 jof-11-00361-t001:** Antifungal susceptibility test results on the *A. nidulans* strain.

Antifungals	MIC (µg/mL)	Breakpoint (µg/mL)		EUCAST Interpretation *
		S≤	>R	
Voriconazole	0.032	IE		NA
Posaconazole	0.064	IE		NA
Itraconazole	0.38	1	2	Susceptible
Isavuconazole	0.047	0.25		Susceptible
Caspofungin	0.19	IE		NA
Amphotericin B	0.5	IE		NA

IE: insufficient evidence; NA = not applicable; * EUCAST interpretation was the version V8 from 2018.

**Table 2 jof-11-00361-t002:** The listing of 13 case reports of spondylodiscitis caused by *A. nidulans*, with findings precisely published, and our case.

Article	Age/Sex	Immune Status	Symptoms	Vertebral Location	Fungal Diagnosis	Surgery	Antifungal Treatment	Outcome
n°1—Redmond et al., 1965 [10]	6/M	IC but previous lung and rib infection	Increasing listlessness, anorexia, weight loss, and breathlessness on exertion	T1–T8	*Postmortem*: hyphal fragments in vertebral lesions + culture in paravertebral pus	No	AMB	Died in few days
n°2—Altman et al., 1977 [11]	10/M	CGD	Incapacitating painful swelling of the right axilla	T2	Culture in the biopsied tissue	Incised and drained axillary abscess	AMB	
			Decreased breath sounds			Excision of a large abscess of the upper lobe of the right lung		Relapsed 3 months later then died
n°3—White et al., 1988 [12]	4/M	CGD	Loss of appetite, weight loss, low-grade fever, and cough	T8–T11	Hyphal elements in histologic sections of lung, vertebral bodies, and ribs	Two debridements	AMB + FC	Survived
					Culture of the paraspinal abscess and the T8, T9, and T11 vertebra		ITRA + FC, then AMB, and then AMB + FC	
n°4—Kim et al., 1997 [13]	6/M	CGD	Low-grade fever and pneumonia	T5–T6	Culture from paraspinal abscess pus	Intensive debridement	AMB + ITRA	Died
n° 5—Segal et al., 1998 [14]	19/M	CGD	Fewer and new left-sided rib pain	T3	Cultures of paraspinal mass at T3	Left upper lobe resection, with partial resection of the third rib	AMB + FC l-AMB	Died
n°6—Segal et al., 1998 [14]	16/M	CGD	Left subcapsular pain and swelling	T4	Hyphae invading the lung and chest wall on hispathologic sections and positive cultures	Left upper lobectomy + resection of the left third and fourth ribs Debridement of the T4 vertebra	AMB, then lc-AMB + ITRA	Survived
n°7—Notheis et al., 2006 [15]	2/M	CGD	Abdominal pain + fever, followed by chest pain and then back pain	T2–T5	Detection of *A. nidulans*, without more precision	Thoracotomy to resect segment 1 of the upper lobe of the lung	d-AMB	Survived (ex vivo gene therapy)
						2nd thoracotomy to resect the right upper lobe	VORI + CASPO, then POSA + CASPO	
						Emergency decompression with laminectomy of T1-T5 vertebrae	l-AMB + POSA + CASPO	
n°8—Dellepiane et al., 2008 [16]	21/M	CGD	Popliteal abscess in the right leg and parenchymal consolidation in the right lung	T5–T7	Culture from the popliteal abscess T5-T7 vertebral body involvement on MRI	Extensive curretages	l-AMB, switched again to VORI + CASPO	Died
n°9—Bukhari et al., 2009 [17]	5/M	CGD	Progressive torticollis, upper back swelling, and weight loss	T1–T2	Hyphae on histopathologic sections; culture with *A. nidulans*	Extensive debridement	VORI for one and half years	Survived
n°10—Jiang et al., 2013 [6]	40/F	IC	Back pain and numbness and weakness of both lower limbs	T1–T3	Fungal granulomatous inflammation on histology	Debridement	VORI followed only for 1 month	Relapsed 16 months later
					16 months later: culture + PCR on biopsy	No surgery at the relapse	VORI for 6 months	Survived
n°11—Lyons et al., 2019 [18]	61/M	IC	Worsening of back pain and lower extremity pain over six weeks	L3–L4	Culture on biopsy Positive galactomannan		VORI for 4 weeks	Relapsed
						Extensive debridement + spinal stabilization	VORI for 7 months + CASPO for 6 weeks	Survived
n°12—Tavakoli et al., 2020 [19]	10/M	CGD	Knee pain, night sweets, lethargy, coryza, and progressive weakness	T4–T5	Surgical debridement samples: filamentous hyphae on histopathology and fungal culture with PCR		AMB then VORI	Relapsed after few months
							VORI + CASPO	
						Insertion of a parietal ventriculoperitoneal shunt + aggressive surgeries	AMB + VORI + CASPO	Died within one year
n°13—present case	59/M	NHL + splenectomy	Back pain for two weeks	L1–L2	Culture on vertebral biopsy Positive galactomannan	No	VORI for 7 months	Survived

M = male, F = female, IC = immunocompetent, CGD = chronic granulomatous disease, NHL = non-Hodgkin lymphoma, AMB = amphotericin B, d-AMB = desoxycholate-AMB, lc-AMB = lipidic complex-AMB, l-AMB = liposomal-AMB, CASPO = caspofungin, ITRA = itraconazole, POSA = posaconazole, VORI = voriconazole, and FC = flucytosine.

## Data Availability

Data used in this study are available upon reasonable request.

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
