# Peer review of "Successful Management of Spondylodiscitis Caused by Aspergillus nidulans: A Case Report and Literature Review"

_jof, 2025, doi:10.3390/jof11050361_

Round 1
Reviewer 1 Report
Comments and Suggestions for Authors
The Case Report of Nadji S et al.. describes the case of SCAN in an immunocompromised patient with non-Hodgkin lymphoma. The filamentous pathogen was identified by microbiology and molecular biology methodologies, and the authors further provided antifungal testing, an important and indispensable result. The report is well-written and important for the scientific community. However, I have some points that should be addressed:
Can the authors provide the ethical approval to publish the data about this case report?
Line 24 - Aspergillus biseriate heads
Figure 1 - there is no A and only B in the first image, please correct.
Line 53 - Aspergillus heads
Line 58 - the ITS4 primer sequence should be written from the 5' to 3' direction
Can the authors provide more details about the PCR (which enzyme was used and the size of the fragment, for example) and the DNA sequencing (where it was performed)?
Lines 61 - 63: Were the galactomannan antigenemia and the (1-3)-β-D-glucan assays performed with the patients' serum? Or LCR? Or other?
Line 69 - please reference the EUCAST breakpoints document
Can the authors provide images after the successful treatment?
The second paragraph of the discussion should be placed in the introduction section (lines 103-107)
Table 2: The authors should include a description of the following abbreviations at the bottom of the table: M, F, IC, and FC.
Line 173: Aspergillus spondylodiscitis
Line 179: (10.8%)
Can the authors discuss the negative test of (1-3)-β-D-glucan assay (Fungitell®)?
Can the authors provide further discussion regarding the age and sex of the patients affected by SCAN? They are mainly male and underage.
Reviewer 2 Report
Comments and Suggestions for Authors
Dear authors,
your clinical case is well presented, and the literature review section is also well done. However, there are some parts that, in my opinion, could be improved and made more fluid to read. I was disappointed not to see images of the isolated fungus.
Successful Management of Spondylodiscitis Caused by Aspergillus nidulans: A Case Report and Literature Review
- 12-13. The sentence ‘emergent pathogen, responsible for various infections 12 but not often involved in osteoarticular infections.’ sounds better if written this way: ‘and emerging pathogen responsible for a variety of infections, though rarely implicated in osteoarticular infections.’
- 14-15. The sentence ‘After an exhaustive diagnostic work-up, he was rapidly and successfully treated with voriconazole’ sounds better if written this way: ‘Following diagnosis, the patient was promptly and effectively treated with voriconazole.’
- 16. Please replace ‘the specific characters’ with ‘the distinctive features’.
- 22. Please replace the sentence with this one ‘In immunocompromised patients, it can lead to sinusitis and pulmonary infections.’
- 24-25. This wording is, in my opinion, more fluid to read ‘which appear brown on the reverse side, its short brown conidiophores, and its biseriate heads.’
- 27. Please replace ‘described fungus’ with ’reported species’
Case report. For improved clarity in understanding the timeline of events, incorporating specific time points, such as 'Day 0', 'Day +30', etc., could enhance readability
- 37. Was the patient considered cured of NHL, or was there any evidence of persistent or relapsed disease?
- 48-49. I assume a complete blood count was performed. Are the leukocyte count and the percentage of neutrophils outside the reference ranges, or is there a specific reason why these values are reported? Would it be possible to include the reference ranges for C-reactive protein as well?
- 51. Bacterial culture. Could you please specify which culture media were used for the bacteriological investigations, as well as the manufacturer of these media?
- 51. Sabouraud plate. Please include the manufacturer.
- 61-64. Regarding the galactomannan antigenemia assay, if I understand correctly, a result of 0.83 with a cut-off value of 1.1 would typically be considered negative, or potentially borderline.
Regarding the (1-3)-beta-D-glucan assay, nclude the interpretation criteria for the test (cut-off or reference interval).
- 64-65. Please replace the sentence with ‘Histopathological examination of the biopsies revealed no evidence of fungal invasion or fungal elements.’
- 66. ‘SCAN was diagnosed’. This sentence is too brief; please provide more context and justification for the diagnosis. For example: Based on the clinical signs, imaging investigations, culture examination, and positive antigenic tests (?), a diagnosis of SCAN was made.
- 66-69. Please replace the sentence with: ‘Antifungal susceptibility test (AFST) was performed using the E-test (bioMer…) to determine the minimum inhibitory concentration (MIC), interpreted according to the breakpoints set by the European Committee on Antimicrobial Susceptibility Testing (EUCAST).’
- 72. Please replace ‘Table 1. : Antifungal susceptibility E-test results on A. nidulans strain’ with ‘Table 1. : Antifungal susceptibility test results on the A. nidulans strain
- 73. ‘Version V8 from 2018.’ The case presented is from 2018. Starting from February 12, 2018, version 9 of the guidelines came into effect. Please verify that the interpretations have not changed compared to version 8 (which, as far as I know, remains the same), which you have considered.
- 75. Add ‘Established’ before ‘breakpoints’.
- 76. Please replace ‘for these’ with ‘Conventionally,’.
- 81-84. To improve the readability, consider better connecting these two sentences, for example, as follows, provided it doesn’t alter the overall meaning: ‘As recommended by the National Reference Center of Mycology, the patient received antifungal treatment with voriconazole for 6 months, with clinical improvement observed after 3 months (from January) and radiologic regression a month later. The center advised discontinuing the therapy based on clinical and imaging findings.’
- 85. Please add the word ‘serum’ before ‘antigen’ (galactomannan serum antigen).
- 87-91. To improve readability, consider merging the sentences. For example: ‘To understand the route of infection, a fibroscopy and bronchoalveolar lavage were performed, revealing an inflammatory aspect of the bronchi. The culture examination from the fibro-aspiration found a colony of A. fumigatus and the development of Candida albicans. Additionally, sinus, cerebral, and pulmonary computer tomography (CT) scans were performed, but no microbiological documentation was found.’
- 94-95. (Thus, SCAN was considered successfully managed. Nevertheless, the patient died from comorbidities 4 years later, in October 2022.) So, there were no relapses of the fungal infection?
- 126. Pleas add the word ‘main’ befor risk (The main risk factors)
- 145-146. In my opinion, this sentence sounds better: ‘A. nidulans was identified through molecular biology, along with persistently positive galactomannan, and susceptibility tests were performed.’
- 149-151. In my opinion, this sentences sounds better: ‘In our case, fibroscopy and bronchoalveolar lavage did not provide contributive findings. Additionally, sinus, cerebral, and pulmonary CT scans revealed no abnormalities. Once again, the route of infection remained unidentified.’
- 156. ‘and vertebral infection decreased’ replace with ‘and vertebral involvement further reduced’.
- 158-160. Please replace the sentence with: ‘The mortality rate of SCAN appears to be high, with potential differences between A. nidulans osteomyelitis and spondylodiscitis, despite the small cohort sizes’.
- 187. Please replace “Of note” with “Notably”.
- 188-189. Please replace ‘…associated a longer treatment with survival and a shorter treatment with increased mortality.’ with ‘…found an association between longer treatment duration and increased survival, whereas shorter treatment was linked to higher mortality.’
- 191. Please replace ‘highlighted the paramount’ with ‘emphasized the critical’
- 192. Please add the word ‘antigungal’ before ‘susceptibility testing’ (antifungal susceptibility testing)
- 205. Please replace‘the contribution of serum galatomannan is still a matter of debate and it is very little used as a diagnostic tool’ with ‘the role of serum galactomannan remains a topic of debate and is rarely used as a diagnostic tool’
- 210-214. Please replace this part with: ‘This case represents the comprehensive management of a SCAN case, detailing the clinical signs, imaging findings, fungal isolation and identification, antifungal susceptibility, and galactomannan antigen dynamics. Based on our current knowledge, it is the first reported case of SCAN in an immunocompromised patient with a malignancy other than CGD to achieve a successful therapeutic outcome without surgical debridement.’
- 227. The correct name is ‘De Hoog GS’.
Comments on the Quality of English Language
I am not a native English speaker, however, some sentences are too short, and in many cases, they could be made more pleasant and fluid by merging them with the following sentences.
Reviewer 3 Report
Comments and Suggestions for Authors
Nadji and colleagues submit a case report and literature review of Aspergillus spondylodiscitis.
Comments:
- Since this course of voriconazole was continued for 6 months, was there any effort to check fluoride levels? (see table 3 of PMID 34644473, under disadvantages of voriconazole, Risk of periostitis or cutaneous cancer with long-term use).
- Were any voriconazole drug levels performed after the original level on day 5 of treatment? What were those results?
- Did liver function tests stay normal throughout treatment?
- Since your case patient had NHL, had splenectomy, and was already immunocompromised enough to be colonized with Aspergillus fumigatus on BAL, should he have been on anti-Aspergillus prophylaxis? See PMID 38481428, Modeling Invasive Aspergillosis Risk for the Application of Prophylaxis Strategies.
- For another review on this topic see PMID 36448782, Osteoarticular Mycoses.
- What is a “control” MRI? Would that just be a second, confirmatory MRI? This use of the term control is unfamiliar to the reviewer.
- In table 1, should the two instances of the word “sensible” be changed to “sensitive”?
- “was done in January 2024 and” does not need to be in italics.
Round 2
Reviewer 1 Report
Comments and Suggestions for Authors
The authors should keep in mind that, although this is a case report, the title of the manuscript describes it as both a case report and a literature review. To write an appropriate paper for the scientific community, where people with different backgrounds will read it, some critical information should be included.
The introduction is incomplete; more information about the disease and fungal infection should be added.
If the authors do not intend to discuss the age and sex of the patients affected by SCAN, as ultimately corresponds to the characteristics of patients with CGD, this must be stated in the introduction.
If the authors do not wish to discuss the negative 1-3-β-D-glucan assay test result, it should not be included in the case report, as it is not relevant. If it is relevant, it must be discussed in more detail.
The authors must include references to the EUCAST methodology and EUCAST breakpoints in the reference list.
The authors should specify in the methodology section where the sequencing was performed and which enzyme was used to amplify the ITS DNA sequence. If a high fidelity enzyme was not used, this could compromised the fungal identification.
Author Response
Reviewer 1 comments
The authors should keep in mind that, although this is a case report, the title of the manuscript describes it as both a case report and a literature review. To write an appropriate paper for the scientific community, where people with different backgrounds will read it, some critical information should be included.
Q. The introduction is incomplete; more information about the disease and fungal infection should be added.
A. I am surprised by the new comments from reviewer #1. In their previous comments, they had highlighted that the case was well-written and important for the scientific community.
Q. If the authors do not intend to discuss the age and sex of the patients affected by SCAN, as ultimately corresponds to the characteristics of patients with CGD, this must be stated in the introduction.
A. Following this comment, we have added a sentence in the discussion regarding age and gender in relation to CGD complications. The sentence is highlighted in orange in the discussion.
Q. If the authors do not wish to discuss the negative 1-3-β-D-glucan assay test result, it should not be included in the case report, as it is not relevant. If it is relevant, it must be discussed in more detail.
A. The limitation of the (1-3)-β-D-glucan assay has been added to the discussion in the form of a sentence highlighted in orange.
Q. The authors must include references to the EUCAST methodology and EUCAST breakpoints in the reference list.
A. We have chosen to present the EUCAST breakpoints used in Table 1, along with the EUCAST version used.
Q. The authors should specify in the methodology section where the sequencing was performed and which enzyme was used to amplify the ITS DNA sequence. If a high fidelity enzyme was not used, this could compromised the fungal identification.
A. We thought we had addressed the methodological aspects in our first response. A highly accurate enzyme was used.
Reviewer 3 Report
Comments and Suggestions for Authors
What does the word "reference" mean here: "E-test (bioMerieux®, reference)"
Author Response
Reviewer 3 comments
What does the word "reference" mean here: "E-test (bioMerieux®, reference)".
Yes, for clarity, only the supplier is retained.
Round 3
Reviewer 1 Report
Comments and Suggestions for Authors
The manuscript is well-written and important for the scientific community. In my first review, I strongly suggested including more information in the introduction section, which was promptly added following the review of the other reviewer as well. Since the authors chose not to address some of my concerns in the discussion section, I reiterated my request to provide more information in the introduction. In the second revision, the authors chose to address the points raised in my first review, and as a result, the introduction section is now complete.
Additionally, 'A highly accurate enzyme was used' is not scientifically appropriate.
Author Response
please find the attach file below
